# Antibacterial and Anti-Biofilm Efficacy of Endolysin LysAB1245 against a Panel of Important Pathogens

**DOI:** 10.3390/ph17020155

**Published:** 2024-01-25

**Authors:** Rosesathorn Soontarach, Potjanee Srimanote, Supayang Piyawan Voravuthikunchai, Sarunyou Chusri

**Affiliations:** 1Center of Antimicrobial Biomaterial Innovation-Southeast Asia, Faculty of Science, Prince of Songkla University, Songkhla 90110, Thailand; rosesathron_pim@hotmail.com (R.S.); supayang.v@psu.ac.th (S.P.V.); 2Division of Infectious Diseases, Department of Internal Medicine, Faculty of Medicine, Prince of Songkla University, Songkhla 90110, Thailand; 3Graduate in Biomedical Sciences, Faculty of Allied Health Sciences, Thammasat University, Pathum Thani 12121, Thailand; psrimanote01@yahoo.com.au

**Keywords:** antibiotic-resistant bacteria, endolysin, antibacterial agent, anti-biofilm agent

## Abstract

Infections caused by antibiotic-resistant bacteria pose a significant global challenge. This study explores the antibacterial effects of a bacteriophage-derived endolysin, LysAB1245, against important pathogens, including *Acinetobacter baumannii*, *Escherichia coli*, *Klebsiella pneumoniae*, *Pseudomonas aeruginosa,* and *Staphylococcus aureus.* We determined the minimal inhibitory concentration (MIC) and minimal bactericidal concentration (MBC) for all tested isolates. A time–kill study was conducted to evaluate the reduction in bacterial survival following treatment with LysAB1245. Additionally, the effects of LysAB1245 on *P. aeruginosa* K1455 and methicillin-resistant *S. aureus* (MRSA) NPRC 001R-formed biofilms were investigated. The MIC and MBC of LysAB1245 against all the tested isolates ranged from 4.68 to 9.36 µg/mL and 4.68 to 18.72 µg/mL, respectively. The time–kill study demonstrated more than a 4 log CFU/mL (99.99%) reduction in bacterial survival within 6 h of LysAB1245 treatment at 2MIC. LysAB1245 (1/8–1/2MIC) treatment significantly reduced biofilms formed by *P. aeruginosa* and MRSA in a concentration-dependent manner. Furthermore, scanning electron and confocal laser scanning microscopy confirmed the potential inhibition effects on 3-day established biofilms formed on abiotic surfaces upon treatment with LysAB1245 at 2MIC. The findings indicate that endolysin LysAB1245 could be employed as a new alternative therapeutic antibacterial and anti-biofilm agent for combating biofilm-related infections.

## 1. Introduction

The emergence of antimicrobial-resistant pathogens has become one of the most pressing public health threats [1]. Infections caused by these bacteria are often associated with inappropriate and insufficient use of broad-spectrum antibiotics, leading to elevated morbidity and mortality rates, prolonged hospitalization, and increased healthcare costs [2,3]. Gram-positive and Gram-negative bacteria with multiple antibiotic resistances pose a considerable risk for hospital-acquired infections. In 2017, strains of *Staphylococcus aureus*, *Klebsiella pneumoniae*, *Acinetobacter baumannii*, and *Pseudomonas aeruginosa* were designated as global priority pathogens, necessitating urgent research and the development of new antibiotics [4]. These pathogens exhibit a high degree of antibiotic resistance through various mechanisms [5].

Biofilms represent a crucial virulence factor in microbial survival under extreme environmental conditions and severe chronic bacterial infections [6,7]. The structural composition of polysaccharides and cellular polymeric substances in biofilms provides an effective diffusion barrier, limiting the penetration of antibacterial agents [8,9]. Numerous studies have demonstrated the heightened antibiotic resistance of bacteria associated with a biofilm-producing phenotype [10,11,12]. Given the escalating antibiotic resistance among pathogenic bacteria, bacteriophage-encoded endolysins (lysins) are emerging as promising treatment modalities featuring alternative antibacterial agents [13,14,15,16,17,18].

Endolysins are phage-encoded peptidoglycan-degrading enzymes typically used by most dsDNA bacteriophages to lyse the peptidoglycan layer of their hosts and facilitate progeny release at the end of lytic cycle. Recently, novel lytic phages against multidrug-resistant *A. baumannii* strains were isolated from sewage water [19]. The study suggested the phages encoding important enzymes, including endolysin, could serve as potential alternative antibacterial agents against *A. baumannii* infections [19]. Subsequently, a novel phage endolysin, LysAB1245, was successfully expressed, purified, and characterized [20]. This study demonstrated the lytic activity of LysAB1245 against *A. baumannii* isolates with distinct capsular types; however, its activity in other species has not been explored. We hypothesized that exploring the lytic activity of LysAB1245 in other closely related species could enhance the significance of these findings. Therefore, this study aimed to evaluate the antibacterial and antibiofilm effects of LysAB1245 on a panel of pathogenic Gram-negative and Gram-positive bacteria.

## 2. Results and Discussion

### 2.1. Antibacterial Activity Assays

Minimal inhibitory concentration (MIC) and minimal bactericidal concentration (MBC) values of LysAB1245 are presented in Table 1. Endolysin LysAB1245 exhibited broad-spectrum antibacterial effects, with MIC values ranging from 4.68 to 9.36 µg/mL. LysAB1245 exhibited bactericidal activity at MBC range of 4.68 to 18.72 µg/mL. The results revealed that LysAB1245 from a Gram-negative background exhibited potent broad-spectrum antibacterial effects on various Gram-negative and Gram-positive bacteria. In a previous study, endolysin HY-133 displayed MIC50/90 and MBC50/90 values ranging from 0.12 to 0.5 mg/L against methicillin-susceptible *S. aureus* and MRSA isolates [21]. The MIC and MBC values of endolysin LysSS against reference strains of common hospital pathogens, including *A. baumannii*, *K. pneumoniae*, *P. aeruginosa*, *E. coli*, and *S. aureus*, ranged from 0.25 to more than 2 mg/mL [22]. Although our study, consistent with these previous studies, revealed the efficacy of an endolysin against different bacterial strains, the variation in the MIC and MBC values could be attributed to the variations in the secondary and tertiary conformation of the protein and antibacterial efficacy resulting from the unique structural domains of LysAB1245 [20].

### 2.2. Time–Kill Assay

The time–kill study was performed to examine the bactericidal kinetics of LysAB1245 on *P. aeruginosa* (PA01 and K1455) and *S. aureus* (MRSA NPRC 001R and ATCC 25923) strains. The results revealed that LysAB1245 showed the concentration-dependent killing of all tested strains (Figure 1). LysAB1245 at 2MIC (9.36 µg/mL) reduced the bacterial viability of both *P. aeruginosa* PA01 and K1455 strains by more than 4 log CFU/mL or 99.99% within 4 h (Figure 1A,B). In addition, LysAB1245 at 2MIC (18.72 µg/mL) exhibited bactericidal effects on MRSA R001, resulting in an approximately 4 log CFU/mL decrease in the number of viable cells within 6 h (Figure 1C). For *S. aureus* ATCC 25923, a reduction of over 5 log CFU/mL in the viable bacterial count was observed after 4 h of incubation with LysAB1245 at 2MIC (9.36 µg/mL) (Figure 1D).

In this study, two virulent bacterial strains, including *P. aeruginosa* with an overexpressed MexAB-OprM efflux pump (K1455) and MRSA with mecA gene (NPRC 001R), were used to investigate the bactericidal kinetics of LysAB1245. According to a previous study, the N-terminal catalytic domain of phage endolysin LysAB1245 comprises glycosidase domains (N-acetyl-β-D-muramidases) [20]. LysAB1245 disrupts the peptidoglycan cell wall, leading to cell lysis by cleaving the β-1,4 glycosidic bonds between glycan strands [23]. Similar to endolysin SPN9CC, LysAB1245 exhibited exogenous bacteriolytic activity against Gram-negative bacteria without pretreatment with outer membrane permeabilization [24]. These findings suggest that endolysin LysAB1245 could be employed as a potential therapeutic agent effective against both pathogenic Gram-negative and Gram-positive bacteria.

Previous studies have reported that even after several cycles of repeated exposure to phage endolysin at 1/2MIC or MIC, no resistance development was observed in *S. aureus* strains [25,26]. Moreover, endolysin LysAB1245 was derived from a lytic phage T1245. Together, these studies suggest that, within the host strain, endolysin is consistently expressed, and host cells are continually exposed to endolysin without compromising their susceptibility to it. Consequently, the likelihood of bacterial cells developing resistance against endolysin is deemed improbable or occurs at extremely low frequencies.

### 2.3. Effects of LysAB1245 on A. baumannii Cell Surface by Scanning Electron Microscope

The morphology of *A. baumannii* cells after exposure to LysAB1245 at subinhibitory concentration was visualized using a scanning electron microscope (SEM). The scanning electron micrographs demonstrated shrinking deformation of the LysAB1245-exposed cells, whereas untreated cells displayed normal rod-shaped morphology with smooth and intact surfaces (Figure 2). Together with previous studies, our study suggests that a lysozyme-like catalytic domain of bacteriophage endolysin is pivotal in disrupting the integrity of the bacterial outer membrane, resulting in the leakage of cytoplasmic content [27,28].

### 2.4. Inhibition of Biofilm Formation

Biofilms are a survival strategy of several microorganisms against host defense mechanisms or stressed conditions. The biofilm-forming ability of all the clinically tested isolates was investigated using crystal violet staining. The results revealed that all strains of *P. aeruginosa* evaluated in this study were strong biofilm-producers. The pathogenic isolates of *P. aeruginosa* are commonly used as model organisms to study the formation of biofilm due to their potent biofilm-producing capacity [29]. Therefore, MexAB-OprM overexpressed strain K1455, which is a high biofilm-producing strain of *P. aeruginosa*, was selected as a representative strain to observe the inhibitory effects of LysAB1245 on bacterial biofilm formation. Furthermore, MRSA NPRC 001R, a moderate level of biofilm-forming ability, was also selected as a representative Gram-positive bacterium to assess the anti-biofilm efficacy of LysAB1245.

Endolysin is a bacteriophage-associated peptidoglycan hydrolase with the ability to prevent bacterial biofilm formation. This study revealed that the anti-biofilm effects of LysAB1245 against biofilm formed by *P. aeruginosa* and MRSA strains were concentration-dependent. Furthermore, the crystal violet assay demonstrated a significant reduction in biofilm biomass with a maximum reduction at 1/2MIC of LysAB1245 (*p* < 0.0001; Figure 3). For *P. aeruginosa* strain K1455, treatment with LysAB1245 at concentrations of 1/2MIC, 1/4MIC, 1/8MIC, 1/16MIC, and 1/32MIC led to biofilm reductions of approximately 80.54 ± 6.43%, 67.65 ± 2.0%, 52.91 ± 4.22%, 34.81 ± 10.64%, and 20.05 ± 7.45%, respectively, compared with untreated biofilms (Figure 3A). Similarly, in MRSA NPRC 001R, LysAB1245 treatment (at 1/2MIC, 1/4MIC, 1/8MIC, 1/16MIC, and 1/32MIC) led to biofilm reductions of approximately 76.48 ± 2.89%, 62.23 ± 3.55%, 48.70 ± 5.47%, 33.25 ± 9.02%, and 27.87 ± 4.82%, respectively, compared with untreated biofilms (Figure 3B).

### 2.5. Effects of LysAB1245 on Established Biofilm Biomass and Viability of Biofilm Cells

After the formation of a biofilm, bacterial cells exhibit remarkable resistance to various antibiotics. Therefore, the effects of LysAB1245 on 3-day established bacterial biofilms were investigated by measuring biofilm biomass (Figure 4). For *P. aeruginosa* K1455 established biofilms, a significant reduction in biofilm formation was observed, with approximately 5.89 ± 1.81%, 21.03 ± 1.62%, 26.32 ± 1.76%, and 75.26 ± 5.33% of biofilm formation after treatment with LysAB1245 at 8MIC, 4MIC, 2MIC, and MIC, respectively (Figure 4A). Additionally, treatments with LysAB1245 at 8MIC, 4MIC, 2MIC, and MIC significantly reduced established biofilms of MRSA NPRC 001R, with biofilm formation percentages of 11.26 ± 2.56%, 26.14 ± 6.08%, 53.31 ± 2.07%, and 61.06 ± 2.45%, respectively (Figure 4B).

Furthermore, the viability of bacterial cells in biofilms was determined using the MTT colorimetric method (Figure 5). As shown in Figure 5A, treatment with LysAB1245 at concentrations ranging from MIC to 8MIC significantly reduced cell viability of *P. aeruginosa* K1455 and MRSA NPRC 001R 3-day established biofilms in a dose-dependent manner. LysAB1245 at 8MIC exhibited a maximum significant reduction in cell viability (*p* < 0.0001).

Biofilms constitute intricate structures of bacterial communities in which cells are embedded within a self-produced matrix. Bacteria living in a biofilm exhibit greater resistance to antibacterial agents compared to their planktonic cells [30,31,32]. Biofilms formed by *P. aeruginosa* play a significant role in the persistence and endemic spread of nosocomial infections [33,34]. Furthermore, persistent MRSA infections are frequently identified in isolates from patients with cystic fibrosis [35,36]. The formation of biofilms on living tissues or medical devices emerges as a crucial virulence factor in many pathogenic bacteria, leading to chronic infections that significantly complicate the treatment process [37,38]. Due to the association between biofilm formation and antibacterial resistance in many pathogenic bacteria, the prevention and eradication of biofilm formation have become preferred strategies for addressing biofilm-related challenges. This study highlights the importance of the anti-biofilm properties of LysAB1245 in combating bacteria growing in biofilms. The results demonstrated that LysAB1245 induces a significant reduction in both biomass and viability of biofilm cells, established over three days, in both Gram-negative and Gram-positive strains. Therefore, these findings suggest that endolysin LysAB1245 could serve as a promising novel alternative therapeutic approach for tackling biofilm formation and eradication.

### 2.6. Scanning Electron Micrographs of P. aeruginosa K1455 and MRSA NPRC 001R 3-Day Established Biofilms after Exposure to LysAB1245

Scanning electron microscopy was performed to confirm the effects of LysAB1245 on the morphology of established biofilm cells formed on glass slide surfaces. As shown in Figure 6, treatment with LysAB1245 at 2MIC demonstrated effective removal of 3-days established biofilms for both *P. aeruginosa* K1455 and MRSA NPRC 001R strains, compared with untreated biofilms. In addition, cell membrane deformation and poration were observed after exposure to LysAB1245 at 2MIC for 24 h (Figure 7).

In this study, endolysin LysAB1245 demonstrated promising potential as a novel alternative therapeutic for degrading preformed biofilms produced by *P. aeruginosa* K1455 and MRSA NPRC 001R strains on abiotic surfaces. A previous study has shown that treatment with LysSTG2 at 1 mg/mL for 2 h effectively removed over 99.9% of biofilms formed by *P. aeruginosa* and *P. putida* on a stainless-steel surface [39]. Additionally, the activity of endolysin XZ.700 against MRSA biofilms formed on titanium discs was observed, with no cytotoxic effects on human bone cells after 48 h of exposure at 50 µg/mL [40]. Moreover, the lytic activities of endolysin LysH5 have been reported not only against staphylococcal biofilms but also against rifampicin and ciprofloxacin-induced persister cells obtained by treatment with rifampicin [41].

### 2.7. Confocal of P. aeruginosa K1455 3-Day Established Biofilms after Exposure to LysAB1245

A three-dimensional (3D) structural image of live and dead cells within the matrix of established biofilms formed by a representative strain of *P. aeruginosa* K1455 on glass surfaces was visualized using confocal laser scanning microscopy (CLSM; Figure 8). The results revealed that LysAB1245 at 4MIC effectively disrupted preformed biofilms (Figure 8A). At 2MIC, a reduction in cell viability was observed, with live cells fluorescing green (Syto 9 dye) and dead cells stained red (propidium iodide) (Figure 8B). In contrast, untreated biofilms displayed dense aggregations of living bacterial cells with green fluorescence (Figure 8C).

Based on the findings of this study, the phage-derived endolysins LysAB1245 is considered a potential antibacterial agent with antibiofilm properties for controlling both Gram-negative and Gram-positive bacteria. Furthermore, the topical skin application of the phage endolysin formulated as cream has been proven a safe alternative to antibiotic therapy without causing long-term effects and disrupting the normal flora through pre-clinical and clinical trials [42]. In 2017, successful treatment was reported for three cases of atopic dermatitis caused by *S. aureus* infections using long-term daily topical Staphefekt SA.100 therapy [43]. Furthermore, the use of endolysins as effective biocontrol agents against various food-borne pathogens holds promise for future applications in the food industry [39,44,45,46]. However, endolysin LysAB1245 is still in the early stages of development. Considerations should be given to limitations such as pharmacokinetics, route of delivery, and optimal doses.

## 3. Materials and Methods

### 3.1. Bacterial Strains and Culture Conditions

*Pseudomonas aeruginosa* PAO1 strain K767 (wild-type), MexAB-OprM overexpressed strain K1455 (PAO1-nalB), and MexB deletion strain K1523 (PAO1-ΔmexB) were provided by Professor Dr. R. Keith Poole, Queen’s University, Kingston, Ontario, Canada. All clinical isolates, including *Acinetobacter baumannii*, *Escherichia coli, Klebsiella pneumoniae*, and methicillin-resistant *Staphylococcus aureus* (MRSA) with mecA gene, were obtained from Songklanagarind Hospital (Ethical Approval No. REC 59–241–19–6). Reference strains of *A. baumannii* ATCC 19606, *E. coli* ATCC 25922, *K. pneumoniae* ATCC 700603, and *S. aureus* ATCC 25923 were used in this study. The bacteria were cultured in Luria Bertani (LB) (Difco Laboratories, Detroit, MI, USA) at 37 °C for 18 h and maintained in 20% glycerol at −80 °C.

### 3.2. Determination of MIC and MBC

The antibacterial efficacy of endolysin LysAB1245 was determined by standard broth microdilution assay according to Clinical and Laboratory Standard Institute (CLSI) M100-ED33 methods [47]. Briefly, LysAB1245 was two-fold serially diluted in concentrations ranging from 1.05 to 33.68 μg/mL in Mueller Hinton broth (MHB, Difco) in 96-well polystyrene microtiter plates. Subsequently, an equal volume of logarithmic phase bacterial cultures at a density of 1 × 10^6^ CFU/mL was added to each well. After incubation at 37 °C for 18 h, the lowest concentration that inhibited bacterial growth was determined as MIC values. The MBC value is defined as the lowest concentration, which had no bacterial growth on the agar plates. The results were obtained from two independent experiments performed in triplicate.

### 3.3. Kinetics Analysis

A time–kill kinetic assay was performed on two representative Gram-negative and Gram-positive bacteria (*P. aeruginosa* and *S. aureus*). Bacterial cultures at 1 × 10^6^ CFU/mL were added to MHB containing LysAB1245 at 1/2MIC, MIC, and 2MIC and incubated at 37 °C for 2, 4, 6, 8, 12, and 16 h. Samples from each time interval were collected, serially diluted 10-fold, and plated on Mueller Hinton agar to determine the number of log CFU/mL. The results were obtained from two independent experiments performed in triplicate and were represented as mean ± SD (standard deviation).

### 3.4. Effects of LysAB1245 on A. baumannii Cell Surfaces Using Scanning Electron Microscope

The potential effects of LysAB1245 on *A. baumannii* ABMYSP-1245, a primary host for phage T1245, were observed. Briefly, bacterial culture at 1 × 10^6^ CFU/mL was treated with LysAB1245 at 1/2MIC (2.34 μg/mL). The mixture of bacterial cells and MHB served as a control. After incubation at 37 °C for 18 h, the supernatant was discarded by centrifugation at 10,000× *g* for 5 min. The pellets were resuspended with 10 mM phosphate-buffered saline (PBS, pH 7.4), fixed with 2.5% (*w*/*v*) glutaraldehyde (Sigma-Aldrich, St Louis, MO, USA), and dehydrated in a graded series of ethanol. After drying with CO_2_ and coating with gold, the specimens were examined using a HITACHI SU3900 instrument.

### 3.5. Screening of Biofilm Formation

All tested clinical isolates were used to evaluate the biofilm-forming ability as described in a previous study [48]. All bacterial isolates were grown overnight at 37 °C in trypticase soy broth (TSB, Difco) with 0.25% glucose. The cultures were then adjusted to 1 × 10^8^ CFU/mL, inoculated into 96-well microtitre plates (200 μL/well), and incubated at 37 °C for 24 h. The contents of the wells were discarded and gently washed twice with 200 μL PBS. The plates were then dried, stained with 0.1% crystal violet for 30 min, and washed twice with distilled water. After drying of the plates, stained biofilm was dissolved in 100 μL dimethyl sulphoxide (DMSO; Merck, Elkton, VA, USA) and the absorbance was measured at 595 nm using a microplate spectrophotometer. All isolates were assigned into four categories according to the following criteria: optical density at (OD595) < 1, non-biofilm forming; 1 < OD595 < 2, weak; 2 < OD595 < 3, medium; and 3 < OD595, strong.

### 3.6. Effects of LysAB1245 on Biofilm Formation

The anti-biofilm efficacy of LysAB1245 against two representative bacterial strains (*P. aeruginosa* K1455) was investigated [49]. Fifty microliters of bacterial suspensions (at 1 × 10^6^ CFU/mL) were added to the wells containing 50 μL diluted LysAB145 (at 1/32–1/2MIC) or TSB-0.25% glucose (control). After incubation at 37 °C for 18 h, all treatment and control wells were discarded and washed twice with PBS. The amounts of biofilm masses formed on the 96-well plate were measured using the crystal violet staining assay as described in the above method. The results were obtained from two independent experiments performed in triplicate. The following formula was used to calculate the percentage of biofilm formation inhibitory activity:$$Biofilm\;formation\;inhibitory\;activity\;\left(\%\right)=\frac{\left({OD}_{control}-{OD}_{blank}\right)-\left({OD}_{treated}-{OD}_{blank}\right)}{\left({OD}_{control}-{OD}_{blank}\right)}\times 100$$

### 3.7. Established Biofilms Eradication Ability of LysAB1245 and Viability of Biofilm Cells

Established biofilms of *P. aeruginosa* K1455 and MRSA NPRC 001R were grown as described in a previous study with slight modifications [50]. Briefly, 200 μL bacterial suspension in TSB-0.25% glucose at 1 × 10^8^ CFU/mL was added to 96-well plates and incubated at 37 °C. For 3-day biofilms, planktonic cells were removed, and fresh TSB-0.25% glucose was added daily. After 3 days of biofilm growth, non-adherent cells were removed, and biofilms were subsequently treated with 100 μL LysAB1245 at different concentrations (1–8MIC) and incubated at 37 °C for 24 h. To evaluate the ability of LysAB1245 to eradicate established bacterial biofilms, crystal violet staining was employed as described in the above method.

To analyze the bacterial cell viability in biofilms, the MTT reduction assay was performed [38]. Briefly, the biofilms were treated with LysAB1245 for 24 h, and the contents of the wells were removed, replaced with 100 μL MTT solution (0.5 mg/mL in PBS; Sigma-Aldrich), and incubated at 37 °C for 2 h. DMSO was used to solubilize the insoluble formazan crystals, and the cell viability was assessed by measuring OD570 using a microplate spectrophotometer. The results were obtained from two independent experiments performed in triplicate. The following formula was used to calculate the percentage of biofilm formation and viability of biofilm cells:$$Biofilm\;formation\;and\;viability\;of\;biofilm\;cells\;\left(\%\right)=\frac{\left({OD}_{treated}-{OD}_{blank}\right)}{\left({OD}_{control}-{OD}_{blank}\right)}\times 100$$

### 3.8. Scanning Electron Micrographs of P. aeruginosa K1455 3-Day Established Biofilms after Exposure to LysAB1245

The effects of LysAB1245 on established biofilms formed by *P. aeruginosa* K1455 and MRSA NPRC 001R were examined on glass slide surfaces following a previously described method with slight modifications [51]. Briefly, 1 mL of bacterial suspension (at 1 × 10^8^ CFU/mL) was added to 24 wells, with each well containing a glass piece (1 × 1 cm). After incubation for 3 days, the biofilms on glass slides were treated with 500 µL LysAB1245 (at 2MIC) and further incubated at 37 °C for 24 h. The established biofilms on glass slides were incubated with 2.5% glutaraldehyde (Sigma-Aldrich) for 2 h at room temperature. Samples were dehydrated in a graded series of ethanol (20–99.9%), mounted on aluminum stubs, critical point dried, coated with gold, and examined by SEM (Quanta 400 FEG; FEI).

### 3.9. Confocal Laser Scanning Microscope (CLSM)

The effects of LysAB1245 on 3D visualization of biofilm structure were monitored using CLSM. Established biofilms of *P. aeruginosa* K1455 were developed on glass slide surfaces and treated with LysAB1245 (at 2 and 4MIC). The established biofilms on glass slides were stained with a Live/Dead BacLight bacterial viability kit (Invitrogen, Carlsbad, CA, USA) and incubated for 15 min in the dark. The samples were observed under a CLSM (OLYMPUS FV3000).

### 3.10. Statistical Analysis

Statistical analysis was performed using GraphPad Prism v8. Dunnett’s test was used for comparison of means at a 95% confidential interval.

## 4. Conclusions

This study highlights the potential of bacteriophage-derived endolysin LysAB1245 as a versatile antibacterial agent with significant applications in controlling both Gram-negative and Gram-positive bacteria. Treatment with LyAB1245 not only significantly reduced biofilm formation but also eradicated preformed biofilms. In addition, results of SEM and CLSM revealed the disruption of biofilm structure and cell morphology of bacterial 3-day established biofilms on glass slide surfaces. Therefore, LysAB1245 is considered a new promising agent to prevent and control clinically relevant biofilms. However, further investigation into the development of LysAB1245-based therapy for suitable applications should be considered.

## Figures and Tables

**Figure 1 pharmaceuticals-17-00155-f001:**
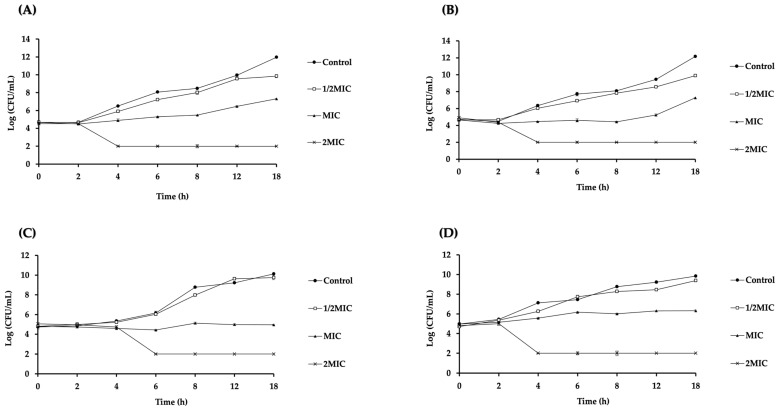
Time–kill curves of Gram-negative and Gram-positive bacteria after treatment with endolysin LysAB1245. Bacterial cultures of *Pseudomonas aeruginosa* PA01 (**A**), *P. aeruginosa* K1455 (**B**), MRSA NPRC 001R (**C**), and *S. aureus* ATCC 25923 (**D**) at 10^6^ CFU/mL were incubated with LysAB1245 at 1/2-2MIC for 18 h. Results obtained from two independent experiments performed in triplicate are expressed as mean ± standard deviation (SD).

**Figure 2 pharmaceuticals-17-00155-f002:**
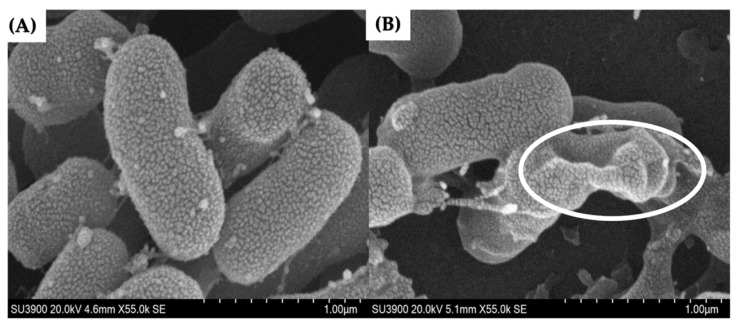
Scanning electron micrographs of untreated (**A**) and LysAB1245-treated *A. baumannii* ABMYH-1245 cells (**B**) at a magnification of 55,000×. The circle indicates the presence of morphological changes.

**Figure 3 pharmaceuticals-17-00155-f003:**
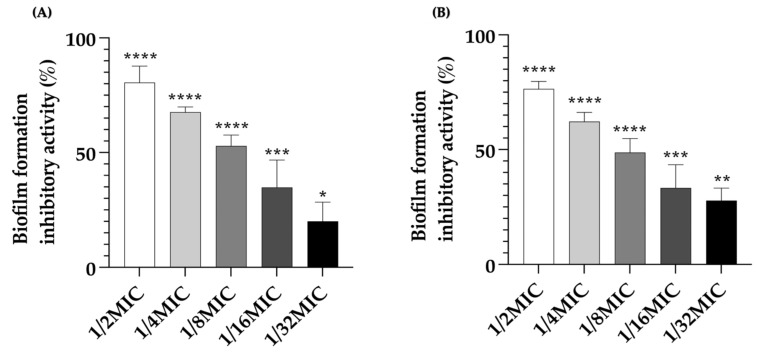
Anti-biofilm effects of LysAB1245 at 1/32-1/2MIC on biofilm formed by *P. aeruginosa* K1455 (**A**) and MRSA NPRC 001R (**B**). Significant reduction in biofilm biomass, compared with control, * *p* < 0.05, ** *p* < 0.01 *** *p* < 0.001, and **** *p* < 0.0001. Results obtained from two independent experiments performed in triplicate are expressed as mean ± standard deviation (SD).

**Figure 4 pharmaceuticals-17-00155-f004:**
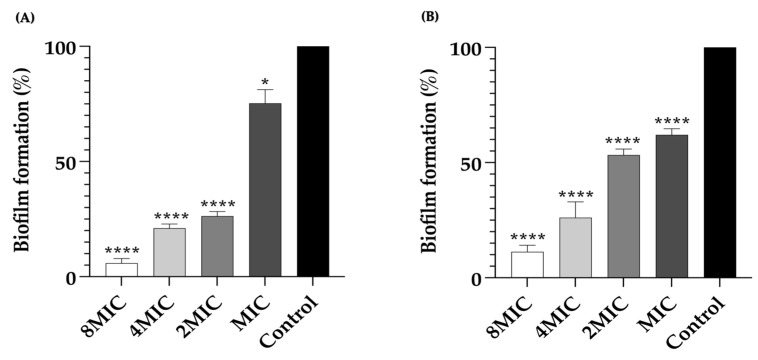
Removal of 3-day established biofilms formed by *P. aeruginosa* K1455 and MRSA NPRC 001R. Biofilms of *P. aeruginosa* K1455 (**A**) and MRSA NPRC 001R (**B**) were treated with LysAB1245 at 1-8MICs for 24 h. Significant reduction in biofilm biomass, compared with control, * *p* < 0.05, and **** *p* < 0.0001. Results obtained from two independent experiments performed in triplicate are expressed as mean ± standard deviation (SD).

**Figure 5 pharmaceuticals-17-00155-f005:**
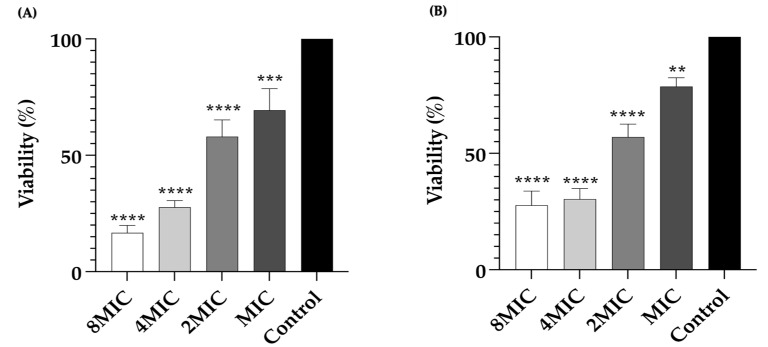
Bacterial viability of 3-day established biofilms formed by *P. aeruginosa* K1455 (**A**) and MRSA NPRC 001R (**B**) after treatment with LysAB1245 at 1–8MICs for 24 h. Significant reduction in biofilm biomass, compared with control, ** *p* < 0.01 *** *p* < 0.001, and **** *p* < 0.0001. Results obtained from two independent experiments performed in triplicate are expressed as mean ± standard deviation (SD).

**Figure 6 pharmaceuticals-17-00155-f006:**
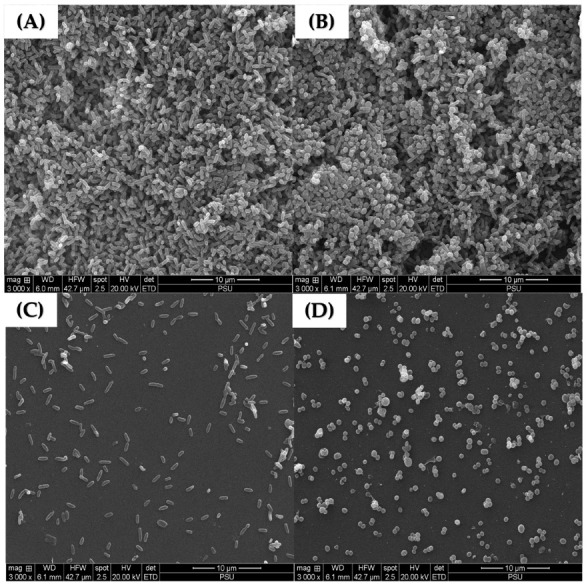
Scanning electron micrographs of 3-day established biofilms on glass slide surfaces at a magnification of 3000×. Scanning electron micrographs of untreated established biofilms formed by *P. aeruginosa* K1455 (**A**) and MRSA NPRC 001R (**B**) present thick bacterial aggregations. Biofilms after treatment with LysAB1245 at 2MIC for 24 h were observed on *P. aeruginosa* K1455 (**C**) and MRSA NPRC 001R (**D**).

**Figure 7 pharmaceuticals-17-00155-f007:**
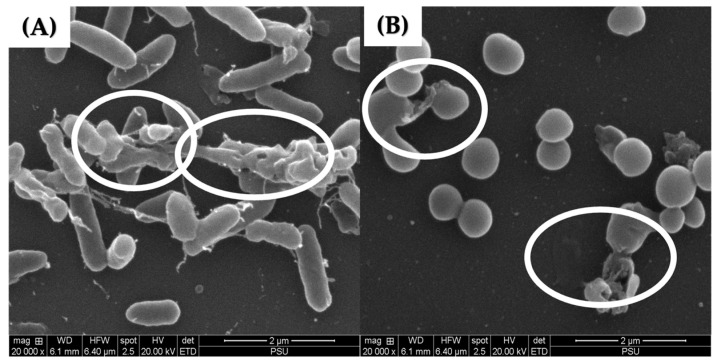
Scanning electron micrographs of 3-day established biofilms formed on glass slide surfaces at a magnification of 20,000×. After treatment with LysAB1245 at 2MIC for 24 h, the destruction of 3-day established biofilms was observed on *P. aeruginosa* K1455 (**A**) and MRSA NPRC 001R (**B**) as indicated by the circle symbol.

**Figure 8 pharmaceuticals-17-00155-f008:**
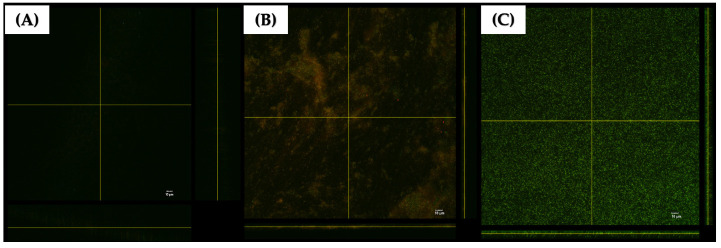
Confocal laser scanning microscopy (CLSM) images of 3-day established biofilms formed by *P. aeruginosa* K1455 on glass slide surfaces. Biofilms were treated with LysAB1245 at 4MIC (**A**), 2MIC (**B**), and control (**C**) for 24 h. All treatments were stained with live/dead staining reagents and subsequent fluorescence microscopy imaging. Living cells are indicated by green Syto 9 staining, and dead bacterial cells are shown in red due to propidium iodide staining.

**Table 1 pharmaceuticals-17-00155-t001:** Minimal inhibitory concentrations (MICs) and minimal bactericidal concentrations (MBCs) of LysAB1245 against pathogenic bacteria.

Pathogenic Organisms	Values of MIC/MBC (μg/mL)
Gram-negative	
*Acinetobacter baumannii*	
NPRCoE 160516	4.68/4.68
ATCC 19606	4.68/4.68
*Escherichia coli*	
NPRCoE 161012	9.36/18.72
ATCC 25922	4.68/9.36
*Klebsiella pneumoniae*	
NPRCoE 160611	9.36/18.72
ATCC 700603	4.68/9.36
*Pseudomonas aeruginosa*	
K767 (PA01)	4.68/9.36
K1455 (PA01-*nalB*)	4.68/9.36
K1523 (PA01-∆*mexB*)	4.68/9.36
Gram-positive	
*Staphylococcus aureus*	
MRSA NPRC 001R	9.36/18.72
ATCC 25923	4.68/9.36

## Data Availability

Data are available in a publicly accessible repository and within the article.

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
