# Peer review of "Antibacterial and Anti-Biofilm Efficacy of Endolysin LysAB1245 against a Panel of Important Pathogens"

_pharmaceuticals, 2024, doi:10.3390/ph17020155_

Round 1

Reviewer 1 Report

Comments and Suggestions for Authors

Thank you for the opportunity to review the manuscript. The work is written in a correct and understandable manner. Below, I provide several observations:

98 - not "from two independent replicates" but maybe "with..."

149 - the experiments are performed in triplicates, so it would be better to present data as % +/- S.D. (or data ranges)

157 - data are presented, not provided

176 - this figure is hard to interpret because of overlaying of panels. 

Generally, I don't understand what does mean in most of figure caption: "Data represent the mean ± SD of three experiments from two

independent repeats." Has it been performed 2 independent repetitions from which 3 experiments were taken into account to calculate SD? It doesn't make sense, please clarify it.

On the other hand, in Materials and Methods section the Authors write "Two independent experiments were performed in triplicate". Please, unify this.

Comments on the Quality of English Language

Some language corretions are needed. I have explained it in my comments to the Authors.

Author Response

Thank you very much for your kind consideration for this manuscript. We have carefully edited the manuscript as suggested by reviewers.

98 - not "from two independent replicates" but maybe "with..."

The authors have edited throughout this manuscript.

149 - the experiments are performed in triplicates, so it would be better to present data as % +/- S.D. (or data ranges)

Information added (Page 5, Line 164, 167, 179) (Page 6, Line 186).

157 - data are presented, not provided

The authors have edited throughout this manuscript.

176 - this figure is hard to interpret because of overlaying of panels. 

Figure 4 was edited (Please see the attachment).

Generally, I don't understand what does mean in most of figure caption: "Data represent the mean ± SD of three experiments from two independent repeats." Has it been performed 2 independent repetitions from which 3 experiments were taken into account to calculate SD? It doesn't make sense, please clarify it. On the other hand, in Materials and Methods section the Authors write "Two independent experiments were performed in triplicate". Please, unify this.

Two independent experiments were performed (each performed in triplicate) with similar results and one representative replicate is shown.

Some language corretions are needed. I have explained it in my comments to the Authors.

This manuscript was extensively checked by Editage to improve the use of English (Please see the attachment).

Reviewer 2 Report

Comments and Suggestions for Authors

Thank you very much for allowing me to review this manuscript. It presents encouraging results regarding the treatment of multi-resistant bacteria, as well as the possibility of controlling their growth in biofilms, a significant risk factor in a large number of infections, especially at the nosocomial level.

From my perspective, the article, in its current format, meets the quality criteria to be published in the Pharmaceuticals journal. I would like to make a few comments solely with the aim of contributing to the improvement of the format:

  1. Abstract: Well-structured and developed. The only thing I would suggest is that the authors remove the word 'potent' from the text (line 13). Describing 'potent antibacterial effects' implies a value judgment that seems not yet proven. This word could be used in the conclusions, but not in the abstract, as it introduces a subjective view to the reader.

  2. Throughout the text, figure captions appear far from the figures they refer to. For example, in lines 94-98, Figure 1 appears several paragraphs below where the figure is inserted. The same issue occurs with the text for Figure 2 (lines 127-129).

Congratulations to the authors for this excellent work.

Author Response

Thank you very much for your kind consideration for this manuscript. We have carefully edited the manuscript as suggested by reviewers.

Abstract: Well-structured and developed. The only thing I would suggest is that the authors remove the word 'potent' from the text (line 13). Describing 'potent antibacterial effects' implies a value judgment that seems not yet proven. This word could be used in the conclusions, but not in the abstract, as it introduces a subjective view to the reader.

Deleted (Page 1, Line 13)

Throughout the text, figure captions appear far from the figures they refer to. For example, in lines 94-98, Figure 1 appears several paragraphs below where the figure is inserted.

The same issue occurs with the text for Figure 2 (lines 127-129).

Edited throughout this manuscript (Please see the attachment).
